# SPARSE NETWORKS FROM SCRATCH:
# FASTER TRAINING WITHOUT LOSING PERFORMANCE

## ABSTRACT

We demonstrate the possibility of what we call sparse learning: accelerated training of deep neural networks that maintain sparse weights throughout training while achieving dense performance levels. We accomplish this by developing sparse momentum, an algorithm which uses exponentially smoothed gradients (momentum) to identify layers and weights which reduce the error efficiently. Sparse momentum redistributes pruned weights across layers according to the mean momentum magnitude of each layer. Within a layer, sparse momentum grows weights according to the momentum magnitude of zero-valued weights. We demonstrate state-of-the-art sparse performance on MNIST, CIFAR-10, and ImageNet, decreasing the mean error by a relative 8%, 15%, and 6% compared to other sparse algorithms. Furthermore, we show that sparse momentum reliably reproduces dense performance levels while providing up to 5.61x faster training. In our analysis, ablations show that the benefits of momentum redistribution and growth increase with the depth and size of the network.

## 1 INTRODUCTION

Current state-of-the-art neural networks need extensive computational resources to be trained and can have capacities of close to one billion connections between neurons (Vaswani et al., 2017; Devlin et al., 2018; Child et al., 2019). One solution that nature found to improve neural network scaling is to use sparsity: the more neurons a brain has, the fewer connections neurons make with each other (Herculano-Houzel et al., 2010). Similarly, for deep neural networks, it has been shown that sparse weight configurations exist which train faster and achieve the same errors as dense networks (Frankle and Carbin, 2019). However, currently, these sparse configurations are found by starting from a dense network, which is pruned and re-trained repeatedly – an expensive procedure.

In this work, we demonstrate the possibility of training sparse networks that rival the performance of their dense counterparts with a single training run – no re-training is required. We start with random initializations and maintain sparse weights throughout training while also speeding up the overall training time. We achieve this by developing sparse momentum, an algorithm which uses the exponentially smoothed gradient of network weights (momentum) as a measure of persistent errors to identify which layers are most efficient at reducing the error and which missing connections between neurons would reduce the error the most. Sparse momentum follows a cycle of (1) pruning weights with small magnitude, (2) redistributing weights across layers according to the mean momentum magnitude of existing weights, and (3) growing new weights to fill in missing connections which have the highest momentum magnitude.

We compare the performance of sparse momentum to compression algorithms and recent methods that maintain sparse weights throughout training. We demonstrate state-of-the-art sparse performance on MNIST, CIFAR-10, and ImageNet-1k. For CIFAR-10, we determine the percentage of weights needed to reach dense performance levels and find that AlexNet, VGG16, and Wide Residual Networks need between 35-50%, 5-10%, and 20-30% weights to reach dense performance levels. We also estimate the overall speedups of training our sparse convolutional networks to dense performance levels on CIFAR-10 for optimal sparse convolution algorithms and naive dense convolution algorithms compared to dense baselines. For sparse convolution, we estimate speedups between 2.74x and 5.61x and for dense convolution speedups between 1.07x and 1.36x. In your analysis, ablations demonstrate

that the momentum redistribution and growth components are increasingly important as networks get deeper and larger in size – both are critical for good ImageNet performance.

## 2 RELATED WORK

**From Dense to Sparse Neural Networks**: Work that focuses on creating sparse from dense neural networks has an extensive history. Earlier work focused on pruning via second-order derivatives (LeCun et al., 1989; Karnin, 1990; Hassibi and Stork, 1992) and heuristics which ensure efficient training of networks after pruning (Chauvin, 1988; Mozer and Smolensky, 1988; Ishikawa, 1996). Recent work is often motivated by the memory and computational benefits of sparse models that enable the deployment of deep neural networks on mobile and low-energy devices. A very influential paradigm has been the iterative (1) train-dense, (2) prune, (3) re-train cycle introduced by Han et al. (2015). Extensions to this work include: Compressing recurrent neural networks and other models (Narang et al., 2017; Zhu and Gupta, 2018; Dai et al., 2018), continuous pruning and re-training (Guo et al., 2016), joint loss/pruning-cost optimization (Carreira-Perpinán and Idelbayev, 2018), layer-by-layer pruning (Dong et al., 2017), fast-switching growth-pruning cycles (Dai et al., 2017), and soft weight-sharing (Ullrich et al., 2017). These approaches often involve re-training phases which increase the training time. However, since the main goal of this line of work is a compressed model for mobile devices, it is desirable but not an important main goal to reduce the run-time of these procedures. This is contrary to our motivation. Despite the difference in motivation, we include many of these dense-to-sparse compression methods in our comparisons. Other compression algorithms include $L_0$ regularization (Louizos et al., 2018), and Bayesian methods (Louizos et al., 2017; Molchanov et al., 2017). For further details, see the survey of Gale et al. (2019).

**Interpretation and Analysis of Sparse Neural Networks**: Frankle and Carbin (2019) show that "winning lottery tickets" exist for deep neural networks – sparse initializations which reach similar predictive performance as dense networks and train just as fast. However, finding these winning lottery tickets is computationally expensive and involves multiple prune and re-train cycles starting from a dense network. Followup work concentrated on finding these configurations faster (Frankle et al., 2019; Zhou et al., 2019). In contrast, we reach dense performance levels with a sparse network from random initialization with a single training run while *accelerating* training.

**Sparse Neural Networks Throughout Training**: Methods that maintain sparse weights throughout training through a prune-redistribute-regrowth cycle are most closely related to our work. Bellec et al. (2018) introduce DEEP-R, which takes a Bayesian perspective and performs sampling for prune and regrowth decisions – sampling sparse network configurations from a posterior. While theoretically rigorous, this approach is computationally expensive and challenging to apply to large networks and datasets. Sparse evolutionary training (SET) (Mocanu et al., 2018) simplifies prune-regrowth cycles by using heuristics: (1) prune the smallest and most negative weights, (2) grow new weights in random locations. Unlike our work, where many convolutional channels are empty and can be excluded from computation, growing weights randomly fills most convolutional channels and makes it challenging to harness computational speedups during training without specialized sparse algorithms. SET also does not include the cross-layer redistribution of weights which we find to be critical for good performance, as shown in our ablation study. The most closely related work to ours is Dynamic Sparse Reparameterization (DSR) by Mostafa and Wang (2019), which includes the full prune-redistribute-regrowth cycle. However, DSR requires some specific layers to be dense. Our method works in a fully sparse setting and is thus more generally applicable. More distantly related is Single-shot Network Pruning (SNIP) (Lee et al., 2019), which aims to find the best sparse network from a single pruning decision. The goal of SNIP is simplicity, while our goal is maximizing predictive and run-time performance. In our experiments, we compare against all four methods: DEEP-R, SET, DSR, and SNIP.

## 3 SPARSE LEARNING

We define sparse learning to be the training of deep neural networks which maintain sparsity throughout training while matching the predictive performance of dense neural networks. To achieve this, intuitively, we want to find the weights that reduce the error most effectively. This is challenging since most deep neural network can hold trillions of different combinations of sparse weights. Additionally,

during training, as feature hierarchies are learned, efficient weights might change gradually from shallow to deep layers. How can we find good sparse configurations? In this work, we follow a divide-and-conquer strategy that is guided by computationally efficient heuristics. We divide sparse learning into the following sub-problems which can be tackled independently: (1) pruning weights, (2) redistribution of weights across layers, and (3) regrowing weights, as defined in more detail below.

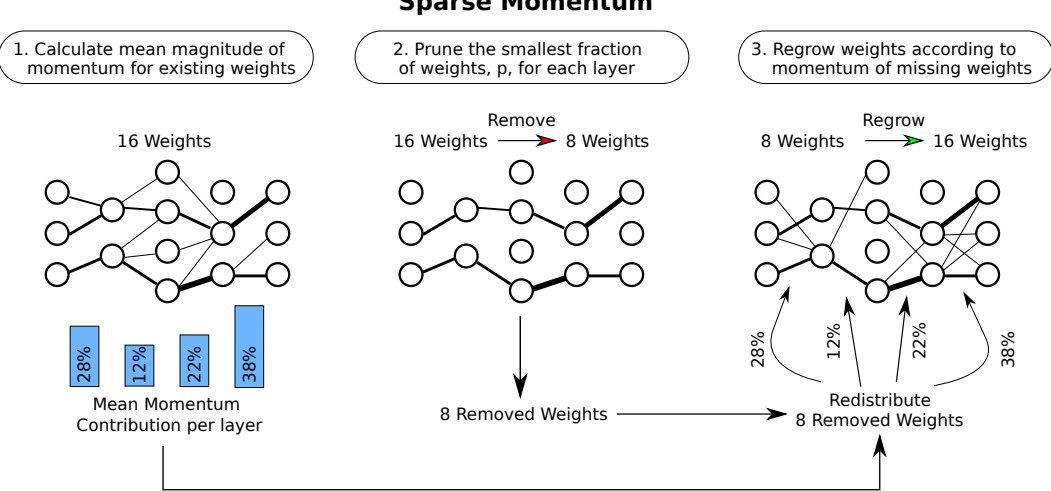

Figure 1: Sparse Momentum is applied at the end of each epoch: (1) take the magnitude of the exponentially smoothed gradient (momentum) of each layer and normalize to 1; (2) for each layer, remove $p = 20\%$ of the weights with the smallest magnitude; (3) across layers, redistribute the removed weights by adding weights to each layer proportionate to the momentum of each layer; within a layer, add weights starting from those with the largest momentum magnitude. Decay $p$.

## 3.1 SPARSE MOMENTUM

We use the mean magnitude of momentum $\mathbf{M}_i$ of existing weights $\mathbf{W}_i$ in each layer $i$ to estimate how efficient the average weight in each layer is at reducing the overall error. Intuitively, we want to take weights from less efficient layers and redistribute them to weight-efficient layers. The sparse momentum algorithm is depicted in Figure 1. In this section, we first describe the intuition behind sparse momentum and then present a more detailed description of the algorithm.

The gradient of the error with respect to a weight $\frac{\partial \mathbf{E}}{\partial \mathbf{W}}$ yields the directions which reduce the error at the highest rate. However, if we use stochastic gradient descent, most weights of $\frac{\partial \mathbf{E}}{\partial \mathbf{W}}$ oscillate between small/large and negative/positive gradients with each mini-batch (Qian, 1999) – a good change for one mini-batch might be a bad change for another. We can reduce oscillations if we take the average gradient over time, thereby finding weights which reduce the error consistently. However, we want to value recent gradients, which are closer to the local minimum, more highly than the distant past. This can be achieved by exponentially smoothing $\frac{\partial \mathbf{E}}{\partial \mathbf{W}}$ – the momentum $\mathbf{M}_i$:

$$\mathbf{M}_i^{t+1} = \alpha \mathbf{M}_i^t + (1 - \alpha)\frac{\partial \mathbf{E}}{\partial \mathbf{W}_i}^t,$$

where $\alpha$ is a smoothing factor, $\mathbf{M}_i$ is the momentum for the weight $\mathbf{W}_i$ in layer $i$; $\mathbf{M}_i$ is initialized at $t = 0$ with $\mathbf{0}$.

Momentum is efficient at accelerating the optimization of deep neural networks by identifying weights which reduce the error consistently. Similarly, the aggregated momentum of weights in each layer should reflect how good each layer is at reducing the error consistently. Additionally, the momentum of zero-valued weights – equivalent to missing weights in sparse networks – can be used to estimate how quickly the error would change if these weights would be included in a sparse network.

The details of the full training procedure of our algorithm are shown in Algorithm 1. See Algorithm 2 in the Appendix for a more detailed, source-code-like description of sparse momentum.

---

**Algorithm 1:** Sparse momentum algorithm.

---

**Data:** Layer i to k with: Momentum $\mathbf{M}_i$, Weight $\mathbf{W_i}$, binary $\mathbf{Mask}_i$ prune rate $p_i$, density $d$

1  **for** $i \leftarrow 0$ **to** $k$ **do**
2      $\mathbf{W}_i \leftarrow$ xavierInit$(\mathbf{W}_i)$
3      $\mathbf{Mask}_i \leftarrow$ createMaskForWeight$(\mathbf{W}_i, d)$
4      applyMask$(\mathbf{W}_i, \mathbf{Mask}_i)$
5  **end**
6  **for** *epoch* $\leftarrow 0$ **to** *numEpochs* **do**
7      **for** $j \leftarrow 0$ **to** *numBatches* **do**
8          batch $\leftarrow$ getBatch$(j)$
9          $\frac{\partial \mathbf{E}}{\partial \mathbf{W}} =$ computeGradients$(\mathbf{W},$ batch$)$
10         UpdateMomentum$(\frac{\partial \mathbf{E}}{\partial \mathbf{W}})$
11         UpdateWeights$(\mathbf{M})$
12         **for** $i \leftarrow 0$ **to** $k$ **do**
13             applyMask$(\mathbf{W}_i, \mathbf{Mask}_i)$
14         **end**
15     **end**
16     totalMomentum $\leftarrow$ getTotalMomentum$(\mathbf{M})$
17     totalPruned $\leftarrow$ getTotalPrunedWeights$(\mathbf{W}, p)$
18     **for** $i \leftarrow 0$ **to** $k$ **do**
19         $m_i \leftarrow$ getMomentumContribution$(\mathbf{M}_i, \mathbf{Mask}_i,$ totalMomentum$)$
20         magnitudePruneWeight$(\mathbf{W}_i, \mathbf{Mask}_i, p_i)$
21         regrowWeights$(\mathbf{W}_i, \mathbf{Mask}_i, m_i \cdot$ totalPruned$)$
22         $p_i \leftarrow$ decayPrunerate$(p_i)$
23         applyMask$(\mathbf{W}_i, \mathbf{Mask}_i)$
24     **end**
25 **end**

---

Before training, we initialize the network with a certain sparsity $s$: we initialize the network as usual and then remove a fraction of $s$ weights for each layer. We train the network normally and mask the weights after each gradient update to enforce sparsity. We apply sparse momentum after each epoch. We can break the sparse momentum into three major parts: (a) redistribution of weights, (b) pruning weights, (c) regrowing weights. In step (a), we we take the mean of the element-wise momentum magnitude $m_i$ that belongs to all nonzero weights for each layer $i$ and normalize the value by the total momentum magnitude of all layers $\sum_{i=0}^{k} m_i$. The resulting proportion is the momentum magnitude contribution for each layer. The number of weights to be regrow in each layer is the total number of removed weights multiplied by each layers momentum contribution: Regrow$_i =$ Total Removed $\cdot m_i$. In step (b), we prune a proportion of $p$ (prune rate) of the weights with the lowest magnitude for each layer. In step (c), we regrow weights by enabling the gradient flow of zero-valued (missing) weights which have the largest momentum magnitude.

Additionally, there are two edge-cases which we did not include in Algorithm 1 for clarity. (1) If we allocate more weights to be regrown than is possible for a specific layer, for example regrowing 100 weights for a layer of maximum 10 weights, we redistribute the excess number of weights equally among all other layers. (2) For some layers, our algorithm will converge in that the average weight in layer $i$ has much larger momentum magnitude than weights in other layers, but at the same time, this layer is dense and cannot grow further. We do not want to prune weights from such important layers. Thus, for these layers, we reduce the prune rate $p_i$ proportional to the sparsity: $p_i = \min(p, \text{sparsity}_i)$.

After each epoch, we decay the prune rate in Algorithm 1 in the same way learning rates are decayed. We use a cosine decay schedule that anneals the prune rate to zero on the last epoch. See Appendix A.1 for an analysis on how decay schedule and starting prune rate affects training.

## 4 EXPERIMENTAL SETUP

For comparison, we follow three different experimental settings, one from Lee et al. (2019) and two settings follow Mostafa and Wang (2019): For MNIST (LeCun, 1998), we use a batch size of 100, decay the learning rate by a factor of 0.1 every 25000 mini-batches. For CIFAR-10 (Krizhevsky and Hinton, 2009), we use standard data augmentations (horizontal flip, and random crop with reflective padding), a batch size of 128, and decay the learning rate every 30000 mini-batches. We train for 100 and 250 epochs on MNIST and CIFAR-10, use a learning rate of 0.1, stochastic gradient descent with Nesterov momentum of $\alpha = 0.9$, and we use a weight decay of 0.0005. We use a fixed 10% of the training data as the validation set and train on the remaining 90%. We evaluate the test set performance of our models on the last epoch. For all experiments on MNIST and CIFAR-10, we report the standard errors. Our sample size is generally between 10 and 12 experiments per method/architecture/sparsity level with different random seeds for each experiment.

We use the modified network architectures of AlexNet, VGG16, and LeNet-5 as introduced by Lee et al. (2019). We consider two different variations of the experimental setup of Mostafa and Wang (2019) for ImageNet and CIFAR-10. The first follows their procedure closely, in that we run the networks in a partially dense setting where the first convolutional layer and downsampling convolutional layers are dense. Additionally, for CIFAR-10 the last fully connected layer is dense. In the second setting, we compare in a fully sparse setting – no layer is dense at the beginning of training. For the fully sparse setting we increase overall number of weights according to the extra parameters in the dense layers and distribute them equally among the network. The parameters in the dense layers make up 5.63% weights of the ResNet-50 network. We refer to these two settings as the partially dense and fully sparse settings.

On ImageNet (Deng et al., 2009), we use ResNet-50 (He et al., 2016) with a stride of 2 for the 3x3 convolution in the bottleneck layers. We use a batch size of 256, input size of 224, momentum of $\alpha = 0.9$, and weight decay of $10^{-4}$. We train for 100 epochs and report validation set performance after the last epoch. We report results for the fully sparse and the partially dense setting.

For all experiments, we keep biases and batch normalization weights dense. We tuned the prune rate $p$ and momentum rate $\alpha$ searching the parameter space $\{0.2, 0.3, 0.4, 0.5, 0.6, 0.7\}$ and $\{0.5, 0.6, 0.7, 0.8, 0.9, 0.95, 0.99\}$ on MNIST and CIFAR-10 and found that $p = 0.2$ and $\alpha = 0.9$ work well for most architectures. We use this prune and momentum rate throughout all experiments.

ImageNet experiments were run on 4x RTX 2080 Ti and all other experiments on individual GPUs.

Our software builds on PyTorch (Paszke et al., 2017) and is a wrapper for PyTorch neural networks with a modular architecture for growth, redistribution, and pruning algorithms. Currently, no GPU-accelerated libraries that utilize sparse tensors exist, and as such we use masked weights to simulate sparse neural networks. Using our software, any PyTorch neural network can be adapted to be a sparse momentum network with less than 10 lines of code. We will open-source our software along with trained models and individual experimental results.[1]

## 5 RESULTS

Results in Figure 2 and Table 1 show a comparison with model compression methods. On MNIST, sparse momentum is the only method that provides consistent strong performance across both LeNet 300-100 and LeNet-5 Caffe models. Soft-weight sharing (Ullrich et al., 2017) and Layer-wise Brain Damage (Dong et al., 2017) are competitive with sparse momentum for one model, but under-performs for the other model. For 1-2% of weights, variational dropout is more effective – but this method also uses dropout for further regularization while we only use weight decay. We can see that sparse momentum achieves equal performance to the LeNet-5 Caffe dense baseline with 8% weights.

On CIFAR-10 in Table 1, we can see that sparse momentum outperforms Single-shot Network Pruning (SNIP) for all models and can achieve the same performance level as a dense model for VGG16-D with just 5% of weights.

---

[1] https://www.dropbox.com/s/wes0wtt75iad4j4/sparse_learning.zip?dl=0

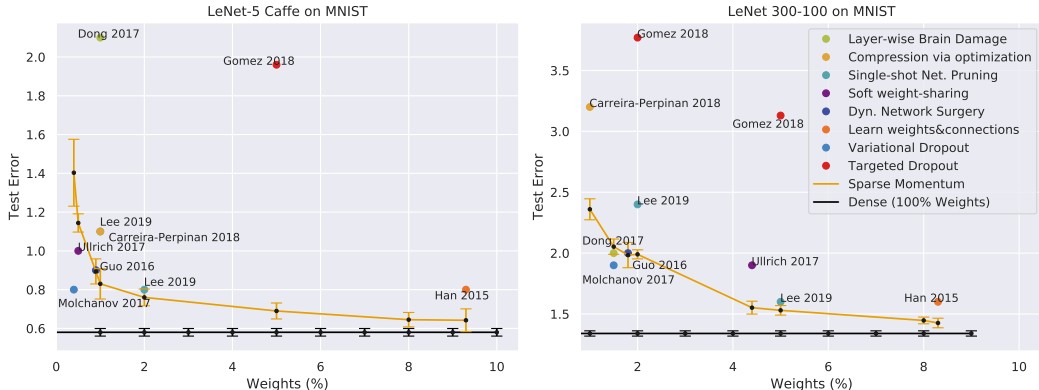

Figure 2: Comparisons against compression methods on MNIST with 95% confidence intervals.

Figure 3 and Table 2 show comparisons of sparse learning methods on MNIST and CIFAR that follows the experimental procedure of Mostafa and Wang (2019) where some selected layers are dense. For LeNet 300-100 on MNIST, we can see that sparse momentum outperforms all other methods. For CIFAR-10, sparse momentum is better than dynamic sparse in 4 out of 5 cases. However, in general, the confidence intervals for most methods overlap – this particular setup for CIFAR-10 with specifically selected dense layers seems to be too easy to determine difference in performance between methods and we do not recommend this setup for future work. Table 2 shows that sparse momentum outperforms all other methods on ImageNet (ILSVRC2012) for the Top-1 accuracy measure. Dynamic sparse is better for the Top-5 accuracy with 20% weights. In the fully sparse setting, sparse momentum remains competitive and seems to find a weight distribution which works equally well for the 10% weights case. For 20% weights, the performance decreases slightly.

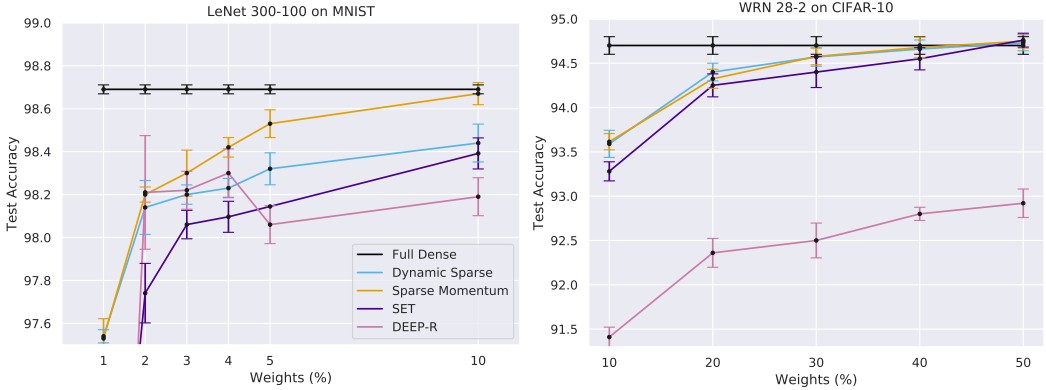

Figure 3: Test set accuracy with 95% confidence intervals on MNIST and CIFAR at varying sparsity levels for LeNet 300-100 and WRN 28-2.

### 5.1 Speedups and Weights Needed for Dense Performance Levels

We analyzed how many weights are needed to achieve dense performance for our networks on CIFAR-10 and how much faster would we able to train such a sparse network compared to a dense one. We do this analysis by increasing the number of weights by 5% until the sparse network trained with sparse momentum reaches a performance level that overlaps with a 95% confidence interval of the dense performance. We then measure the speedup of the model. For each network-density combination we perform ten training runs with different random seeds to calculate the mean test error and its standard error.

To estimated the speedups that could be obtained using sparse momentum for these dense networks we follow two approaches: Theoretical speedups for sparse convolution algorithms which are proportional to reductions in FLOPS and practical speedups using dense convolutional algorithms

Table 1: CIFAR-10 test set error (±standard error) for dense baselines, Sparse Momentum and SNIP.

| Model | Dense Error (%) | Sparse Error (%) | | Weights (%) |
|---|---|---|---|---|
| | | SNIP | Momentum | |
| AlexNet-s | 12.95±0.056 | 14.99 | **14.27**±0.123 | 10 |
| AlexNet-b | 12.85±0.068 | 14.50 | **13.56**±0.094 | 10 |
| VGG16-C | 6.49±0.038 | 7.27 | **7.00**±0.054 | 5 |
| VGG16-D | 6.59±0.050 | 7.09 | **6.69**±0.049[*] | 5 |
| VGG16-like | 6.50±0.054 | 8.00 | **7.00**±0.077 | 3 |
| WRN-16-8 | 4.57±0.022 | 6.63 | **5.62**±0.056 | 5 |
| WRN-16-10 | 4.45±0.040 | 6.43 | **5.24**±0.052 | 5 |
| WRN-22-8 | 4.26±0.032 | 5.85 | **4.93**±0.056 | 5 |

[*] 95% confidence intervals overlap with dense model.

Table 2: Results for ResNet-50 on ImageNet.

| Model | | Accuracy (%) | | | |
|---|---|---|---|---|---|
| | | Top-1 | Top-5 | Top-1 | Top-5 |
| Dense ResNet-50 (He et al., 2016) | | 74.9 | 92.4 | 74.9 | 92.4 |
| | Fully Sparse | 10% weights | | 20% Weights | |
| DeepR (Bellec et al., 2018) | ✗ | 70.2 | 90.0 | 71.7 | 90.6 |
| SET (Mocanu et al., 2018) | ✗ | 70.4 | 90.1 | 72.6 | 91.2 |
| Dynamic Sparse (Mostafa and Wang, 2019) | ✗ | 71.6 | 90.5 | 73.3 | **92.4** |
| Sparse momentum | ✗ | **72.3** | **91.0** | **74.2** | 91.9 |
| | ✓ | **72.3** | **91.0** | 73.8 | 91.8 |

which are proportional to empty convolutional channels. For our sparse convolution estimates, we calculate the FLOPS saved for each convolution operation throughout training as well as the runtime for each convolution. To receive the maximum speedups for sparse convolution, we then scale the runtime for each convolution operation by the FLOPS saved. While a fast sparse convolution algorithm for coarse block structures exist for GPUs (Gray et al., 2017), optimal sparse convolution algorithms for fine-grained patterns do not and need to be developed to enable these speedups.

The second method measures practical speedups that can be obtained with naive, dense convolution algorithms which are available today. Dense convolution is unsuitable for the training of sparse networks but we include this measurement to highlight the algorithmic gap that exists to efficiently train sparse networks. For dense convolution algorithms, we estimate speedups as follows: If a convolutional channel consists entirely of zero-valued weights we can remove these channels from the computation without changing the outputs and obtain speedups. To receive the speedups for dense convolution we scale each convolution operation by the proportion of empty channels. Using these measures, we estimated the speedups for our models on CIFAR-10. The resulting speedups and dense performance levels can be seen in Table 3.

We see that VGG16 networks can achieve dense performance with relatively few weights while AlexNet requires the most weights. Wide Residual Networks need an intermediate level of weights. Despite the large number of weights for AlexNet, sparse momentum still yields large speedups around 3.0x for sparse convolution. Sparse convolution speedups are particularly pronounced for Wide Residual Networks (WRN) with speedups as high as 5.61x. Dense convolution speedups are much lower and are mostly dependent on width, with wider networks receiving larger speedups. These results highlight the importance to develop optimized algorithms for sparse convolution.

Beyond speedups, we also measured the overhead of our sparse momentum procedure to be equivalent of a slowdown to 0.973x±0.029x compared to a dense baseline.

Table 3: Dense performance equivalents and speedups for sparse networks on CIFAR-10.

| Model | Weights (%) | Error(%) | Speedups | |
| --- | --- | --- | --- | --- |
| | | | Dense Convolution (Empty Channels) | Sparse Convolution (FLOPS Reduction) |
| AlexNet-s | 50 | 13.15±0.065 | 1.31x | 3.01x |
| AlexNet-b | 35 | 13.00±0.065 | 1.21x | 2.74x |
| VGG16-C | 10 | 6.64±0.040 | 1.32x | 3.85x |
| VGG16-D | 5 | 6.49±0.045 | 1.36x | 3.51x |
| VGG16-like | 5 | 6.46±0.036 | 1.32x | 3.48x |
| WRN 16-8 | 30 | 4.72±0.051 | 1.07x | 4.59x |
| WRN 16-10 | 25 | 4.56±0.037 | 1.07x | 4.41x |
| WRN 22-8 | 20 | 4.40±0.037 | 1.21x | 5.61x |

# 6 ANALYSIS

## 6.1 ABLATION ANALYSIS

Our method differs from previous methods like SET and Dynamic Sparse Reparameterization in two ways: (1) redistribution of weights and (2) growth of weights. To understand the performance contribution of these components, we perform ablations on CIFAR-10 for VGG16-D with 5% weights, MNIST for LeNet 300-100 and LeNet-5 Caffe with 5% weights, and ImageNet for ResNet-50 with 10% weights in the fully sparse setting. The results can be seen in Table 4.

**Redistribution**: Redistributing weights according to the momentum magnitude becomes increasingly important the larger a network is as can be seen from the steady increases in error from the small LeNet 300-100 to the large ResNet-50 when no momentum redistribution is used. Increased test error is particularly pronounced for ImageNet where the Top-1 error increases by 3.42% to 9.71% if no redistribution is used.

**Momentum growth**: Momentum growth improves performance over random growth by a large margin for ResNet-50 on ImageNet, but for smaller networks the combination of redistribution and random growth seems to be sufficient to find good weights. Random growth without redistribution, however, cannot find good weights. These results suggest that with increasing network size a random search strategy becomes inefficient and smarter growth algorithms are required for good performance.

Table 4: Ablation analysis for different growth and redistribution algorithm combinations for LeNet 300-100 and LeNet-5 Caffe on MNIST, VGG16-D on CIFAR-10, and ResNet-50 on ImageNet.

| Redistribution | Growth | Test error in % | | | |
| --- | --- | --- | --- | --- | --- |
| | | LeNet 300-100 | LeNet-5 Caffe | VGG16-D | ResNet-50 |
| momentum | momentum | 1.53±0.020 | 0.69±0.021 | 6.69±0.049 | 27.07 |
| momentum | random | +0.07±0.022 | −0.05±0.011 | −0.19±0.040 | +7.29 |
| None | momentum | +0.01±0.018 | +0.32±0.071 | +1.54±0.101 | +3.42 |
| None | random | +0.11±0.020 | +0.13±0.013 | +1.49±0.147 | +9.71 |

# 7 CONCLUSION AND FUTURE WORK

We presented our sparse learning algorithm, sparse momentum, which uses the mean magnitude of momentum to grow and redistribute weights. We showed that sparse momentum outperforms other sparse algorithms on MNIST, CIFAR-10, and ImageNet. Additionally, sparse momentum can rival dense neural network performance while accelerating training. Our analysis of speedups highlights the need for research into specialized sparse convolution and sparse matrix multiplication algorithms to enable the benefits of sparse networks.

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

# A   APPENDIX

## A.1   SENSITIVITY ANALYSIS

Sparse momentum depends on two hyperparameters: Prune rate and momentum. In this section, we study the sensitivity of the accuracy of our models as we vary the prune rate and momentum. Since momentum parameter has an additional effect on the optimization procedure, we run control experiments for fully dense networks thus disentangling the difference in accuracy accounted by our sparse momentum procedure.

We run experiments for VGG-D and AlexNet-s with 5% and 10% weights on CIFAR-10. Results can be seen in Figure 4. We see that sparse momentum is highly robust to the choice of prune rate with results barely deviating when the prune rate is in the interval between 0.2 to 0.4. However, we can see a gradual linear trend that indicates that smaller prune rates work slightly better than larger ones. Cosine and linear prune rate annealing schedules do equally well. For momentum, confidence intervals for values between 0.7 and 0.9 overlap indicating that our procedure is robust to the choice of the momentum parameter. Sparse momentum is more sensitive to low momentum values ($\leq 0.6$) while it is less sensitive for large momentum values (0.95) compared to a dense control. Additionally, we test the null hypothesis that sparse momentum is equally sensitive to deviations from a momentum parameter value of 0.9 as a dense control. The normality assumption was violated and data transformations did not help. Thus we use the non-parametric Wilcoxon Signed-rank Test. We find no evidence that sparse momentum is more sensitive to the momentum parameter than a dense control, $W(16) = 22.0, p = 0.58$. Overall, we conclude that sparse momentum is highly robust to deviations of the pruning schedule and the momentum and prune rate parameters.

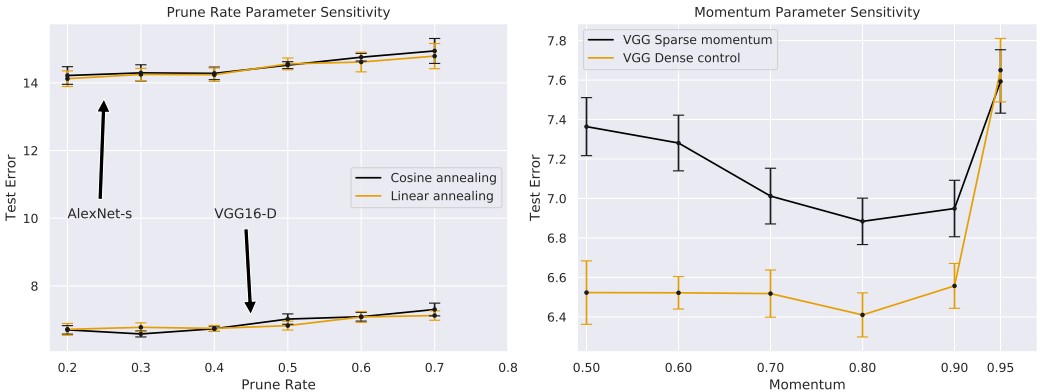

Figure 4: Parameter sensitivity analysis for prune rate and momentum with 95% confidence intervals.

# B   ADDITIONAL ANALYSIS

## B.1   DENSE VS SPARSE FEATURES

Are there differences between feature representations learned by dense and sparse networks? The answer to this question can help with the design of sparse learning algorithms and sparse architectures. In this section, we look at the features of dense and sparse networks and how specialized these features are for certain classes. We test difference between sparse and dense network features statistically.

For feature visualization, it is common to backpropagate activity to the inputs to be able to visualize what these activities represent (Simonyan et al., 2013; Zeiler and Fergus, 2014; Springenberg et al., 2014). However, in our case, we are more interested in the overall distribution of features for each layer within our network, and as such we want to look at the magnitude of the activity in a channel since – unlike feature visualization – we are not just interested in feature detectors but also discriminators. For example, a face detector would induce positive activity for a 'person' class but might produce negative activity for a 'mushroom' class. Both kinds of activity are useful.

With this reasoning, we develop the following convolutional channel-activation analysis: (1) pass the entire training set through the network and aggregate the magnitude of the activation in each convolutional channel separately for each class; (2) normalize across classes to receive for each channel the proportion of activation which is due to each class; (3) look at the maximum proportion of each channel as a measure of class specialization: a maximum proportion of $1/N_c$ where $N_c$ is the number of classes indicates that the channel is equally active for all classes in the training set. The higher the proportion deviates from this value, the more is a channel specialized for a particular class.

We obtain results for AlexNet-s, VGG16-D, and WRN 28-2 on CIFAR-10 and use as many weights as needed to reach dense performance levels. We then test the null hypothesis, that there are no differences in class specialization between features from sparse networks and dense networks. Equal variance assumptions was violated for VGG-D and normality was violated for WRN-28-2, while all assumptions hold for AlexNet-s. For consistency reasons we perform non-parametric Kruskal-Wallis one-way analysis of variance tests for all networks. For AlexNet-s, we find some evidence that features of sparse networks have lower class specialization compared to dense networks $\chi^2(5) = 4.43, p = 0.035$, for VGG-D and WRN-28-2 we find strong evidence that features of sparse networks have lower class specialization than dense networks $\chi^2(13) = 28.1, p < 0.001$, $\chi^2(12) = 36.2, p < 0.001$. Thus we reject the null hypothesis. These results increase our confidence that sparse networks learn features which have lower class specialization than dense networks.

Plots of the distributions of sparse vs. dense features for AlexNet-s, VGG16-D, and WRN 28-2 on CIFAR-10 in Figure 5. These plots were selected to highlight the difference in distribution in the first layers and last layers of each network. We see the convolutional channels in sparse networks have lower class-specialization indicating they learn features which are useful for a broader range of classes compared to dense networks. This trend intensifies with depth.

Overall, we conclude that sparse networks might be able to rival dense networks by learning more general features that have lower class specialization.

## C  FURTHER RESULTS

### C.1  TUNED RESNET-50 ON IMAGENET

We also tried a better version of the ResNet-50 in the fully sparse setting for which we use a cosine learning rate schedule, label smoothing of 0.9, and we warmup the learning rate. The results can be seen in Table 5.

Table 5: Fully sparse ImageNet results.

| Model | Weights (%) | Accuracy (%) | |
|---|---|---|---|
| | | Top-1 | Top-5 |
| Tuned ResNet-50 | 100 | 77.0 | 93.5 |
| Sparse momentum | 10 | 72.9 | 91.5 |
| | 20 | 74.9 | 92.5 |
| | 30 | 75.9 | 92.9 |

## D  DETAILED SPARSE MOMENTUM ALGORITHM

For a detailed NumPy-style algorithmic description of sparse momentum see Algorithm 2.

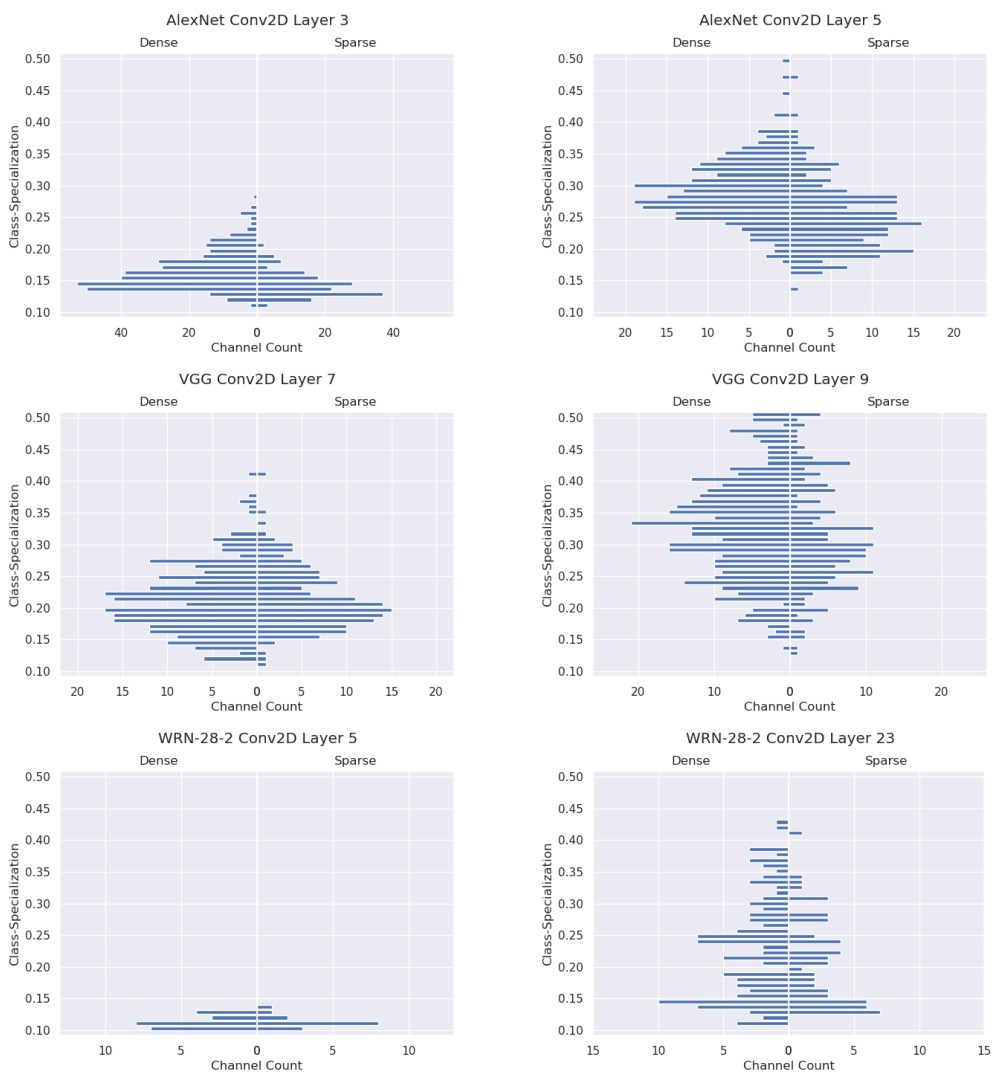

Figure 5: Dense vs sparse histograms of class-specialization for convolutional channels on CIFAR-10. A class-specialization of 0.5 indicates that 50% of the overall activity comes from a single class.

---

**Algorithm 2:** Sparse momentum algorithm in NumPy notation.

---

**Data:** Layer i to k with: Momentum $\mathbf{M}_i$, Weight $\mathbf{W_i}$, binary $\mathbf{Mask}_i$; prune rate $p$

1   TotalMomentum $\leftarrow 0$,   TotalNonzero $\leftarrow 0$

    /* (a) Calculate mean momentum contributions of all layers.     */

2   **for** $i \leftarrow 0$ **to** $k$ **do**

3      MeanMomentum$_i \leftarrow$ mean(abs($\mathbf{M}_i [\mathbf{W}_i \neq 0]$))

4      TotalMomentum $\leftarrow$ TotalMomentum + MeanMomentum$_i$

5      NonZero$_i$ = sum($\mathbf{W}_i \neq 0$)

6      TotalNonzero $\leftarrow$ TotalNonzero + NonZero$_i$

7   **end**

8   **for** $i \leftarrow 0$ **to** $k$ **do**

9      LayerContribution$_i \leftarrow$ MeanMomentum$_i$/TotalMomentum

10     $p_i \leftarrow$ getPruneRate($\mathbf{W}_i, p$)

11     weights by finding the NumRemove*th* smallest weight.

12   **end**

13   **for** $i \leftarrow 0$ **to** $k$ **do**

14     NumRemove$_i \leftarrow$ NonZero$_i \cdot p$

15     PruneThreshold $\leftarrow$ sort(abs($\mathbf{W}_i [\mathbf{W}_i \neq 0]$)) [NumRemove$_i$]

16     $\mathbf{Mask}_i [\mathbf{W}_i <$ PruneThreshold$] \leftarrow 0$ // Stop gradient flow.

17     $\mathbf{W}_i [\mathbf{W}_i <$ PruneThreshold$] \leftarrow 0$

18   **end**

    /* (c) Enable gradient flow of weights with largest momentum

      magnitude.                */

19   **for** $i \leftarrow 0$ **to** $k$ **do**

20     RegrowthThreshold$_i \leftarrow$ sort(abs($\mathbf{M}_i [\mathbf{W}_i == 0]$)) [NumRegrowth$_i$]

21     $\mathbf{Z}_i = \mathbf{M}_i \cdot (\mathbf{W}_i == 0)$ // Only consider the momentum of missing

       weights.

22     $\mathbf{Mask}_i \leftarrow \mathbf{Mask}_i$ | $(\mathbf{Z}_i >$ RegrowthThreshold$_i)$ // | is the boolean OR

       operator

23   **end**

24   $p \leftarrow$ decayPruneRate($p$)

25   applyMask()

---

