# OpenReview forum: "Sparse Networks from Scratch: Faster Training without Losing Performance"
_ICLR.cc/2020/Conference — Reject_

### Official Review · AnonReviewer3 · 2019-10-20
**Official Blind Review #3**

**Rating:** 3

**Review:**


The paper proposes a method to train a sparse network and achieve "dense-level" performance. The method redistributes the sparsity according to momentum contribution of each layer during training after each epoch. Experiments on multiple datasets and architectures are conducted.

While the method itself seems interesting, I do have have several important concerns.

1. My biggest concern is about the experiments. Some results sometime seem questionable.

1) Looking at the results for ImageNet ResNet50, the 20%-weights model achieve 0.7% test error loss compared with dense baseline. According to Table 6 in [1], using the simple pruning and fine-tuning method (Han et al. 2015), the 40%-weights model suffers from 0.06% error loss compared with dense baseline. I don't see a clear advantage of sparse momentum here. It seems possible that under the same sparsity level just pruning using Han et al. is better. Can the authors compare with Han et al. 2015 fairly at the same sparsity level, on multiple datasets? An apple-to-apple comparison is extremely necessary.

Also the baseline result differ: in [1] dense ResNet-50 achieve 76.15% but in this paper baseline result is 74.9%. Both papers use Pytorch. And Pytorch official number is actually 76.15% (see https://pytorch.org/docs/stable/torchvision/models.html ). In my experience that is reproduceble if you follow the official pytorch code of training ImageNet. What's the difference here? I think the authors should follow the standard way of training ImageNet to make the results more convincing.  Finally [1] is a relevant paper and should be discussed.

2) I suggest that MNIST results be moved to Appendix since accuracy on MNIST is too easy to reach a high level, and the interpretation over it might not be convincing. Yet a lot of the analysis of the method's effectiveness is on MNIST.

3) For CIFAR, the results of SNIP on VGG in Table 1 also seems inconsistent with their original paper. In their paper's Table 2, the VGG-like model achieve ~0.3% error reduction at 5% weights while in this paper's Table 1 it's a 0.5% error increase. Again, what is the difference? Is the SNIP method reimplemented? Putting that potentially flawed result aside, the performance on CIFAR is not better than other methods as well according to Figure 3.

2. I don't think the general performance of the trained sparse models can be said to "rival" the dense model. It's a non-negligible margin in most times, for CIFAR and ImageNet. It's a little exaggerating to say that in the title, abstract and introduction.

3. The benefit of the method seems unclear. It does not speedup training compared with the conventional training a dense model and then pruning pipeline (Han et al.). The test-time real speedup is also limited according to Table 3. The real benefit is the compressed model but that is also achievable by traditional pruning (Han et al.).

4. The ablation study should compare with smarter baselines than random regrowth/no redistribution. Clearly each layer needs different level of sparsity so no redistribution is of course not a competitive one. But there might exist other criterion (e.g, weight magnitudes) than momentum which gives good results. The ablation study did not demonstrate why using momentum is justified.

5. "For dense convolution algorithms, we estimate speedups as follows: If a convolutional channel consists entirely of zero-valued weights we can remove these channels from the computation without changing the outputs and obtain speedups." How is it possible that all weights associated with a channel are all pruned, in a sparse pruning setting? A channel typically has at least hundreds or even thousands of weights connected with it, so even with 95% sparsity it's extremely unlikely (consider 0.95^1000). The estimate of speedup might be flawed.

In summary, those serious issues with experiments make me vote a rejection for the paper.

[1] Rethinking the Value of Network Pruning, ICLR 2019.

------------------------------------------------------------------------------------
After author rebuttal:

Thank you for the response.

The authors view that their method is not comparable with Han et al. 15. because this work started training a sparse model while Han et al. 15 trains a dense model and then do pruning. But I'm not sure whether the rather limited actual (1.2-1.3x, and lower on Wide ResNet) speedup justify this argument. What's more, these speedup were said to be "estimated" so I'm not sure whether it's actual speedup.

The pruning ratio with dense performance is not remarkable: 50% sparsity ratio should be easily achievable with Han et al.

Experiments on ResNet (not wide ResNet) were not provided and I assume ResNet would be harder to sparsify than AlexNet, VGG, Wide ResNet which may have more redundancy. I think thus the "faster" training part in the title is not entirely justified.

I'm ok with other parts of the rebuttal and I will raise my score to weak reject. But I still wonder why we should use sparse training in the first place if it brings so little "estimated" speedup.


**Experience Assessment:**

I have published in this field for several years.

**Review Assessment: Checking Correctness Of Derivations And Theory:**

I carefully checked the derivations and theory.

**Review Assessment: Checking Correctness Of Experiments:**

I carefully checked the experiments.

**Review Assessment: Thoroughness In Paper Reading:**

I read the paper thoroughly.

---

> ### Author Response · Authors · 2019-11-05
> **Clarifying information.**
>
> Thank you for your review. Some of your comments indicate that there were some misunderstandings, in particular, note that we train a sparse model from start to finish and thus our method is not directly comparable to Han et al, that speedups are for training and not inference and that dense CIFAR-10 performance is reporting in Table 3. Please see below for detailed comments.
>
> 1.
>
> 1) Our method is not comparable with Han et. al because they start with a dense model while we start with a sparse model. One would assume that our model performs worse because we start, e.g. with 10% weights and keep this budget over the course of training while Han et. al use 100% weights from the beginning and then prune. Thus these methods cannot be directly compared.
>
> 2) We use the same codebase[1] as Mostafa & Wang (2019) to make results comparable to theirs. They reproduce other sparse learning methods like SET and DEEP-R with their codebase as well. Thus running our model on a different codebase would give us an unfair advantage.
> Note, we report an ImageNet baseline that achieves 77.0% error in the appendix to make results comparable across  sub-fields (model pruning vs sparse training), but we think it is more important to make our method comparable to the methods which are closest aligned to our own and thus we would keep the current ImageNet results in the main paper.
>
> 3) We use all the numbers from the SNIP paper[3] except the baseline dense model. They do not mention that they use multiple random seeds for their CIFAR-10 experiments and thus we assume that they only did a single training run (they do not report standard errors). We do 10 experiments with different random seeds and report standard errors. Note that their baseline is within the 99% confidence interval of our baseline. We use the very same codebase that replicated SNIP before[4][5]. So we took all possible imaginable measures to establish equivalence in models: (1) use a ICLR code review codebase that replicates results, (2) verify equivalence using statistical procedures.
>
> 2. We provide dense performance levels on CIFAR-10 in Table 3. We will update the paper emphasizing that dense performance is not achieved in Table 1. It is correct that we do not show that we reach dense performance levels on ImageNet, we run further experiments to establish at which % of weights we reach dense performance levels on ImageNet and report back to you.
>
> 3. Please see Table 3 for _training_ speedups. Note that we speedup training while Han et al. only speeds up inference — our method is not comparable to Han et al. in terms of speedups.
>
> 4. We use the same initialization as SET for our "no-redistribution" ablation. SET does not have any redistribution either and does quite well (see Table 2). Since the main goal of our ablation is to compare performance across sparse training methods this is appropriate.
> We tried weight magnitude, a mix of random + momentum growth, and other procedures which do not offer any benefit. We did not include them in the ablation analysis since they are not comparable to the literature. Our ablation analysis is directly comparable to SET and Dynamic Sparse Reparameterization and thus very relevant. If you still insist on these ablations we will happily produce them, but we doubt they will be useful to understand the performance of our algorithm.
>
> 5. From your comment, I read that you make the assumption that parameters within a convolutional filter are independent. Do you think that is true?
>
> We calculate the channel sparsity in this way (PyTorch):
>
>        sparse_channels = ((layer.weight.data != 0.0).sum([0, 2, 3]) == 0.0).sum().item()
>
> That is we calculate where the feature map is non-zero. Then we sum across the kernel output channels, kernel width, and kernel height which gives us the non-zero elements over the input channels. The channels are empty where the resulting sum is 0.0. Is this calculation incorrect? Note the input x is of dimensionality [out_channel, in_channel, kernel_width, kernel_height].
>
> If you desire, we can also compute speedups using the kernel sparsity. Please let us know so we can run these experiments within the allotted time.
>
> [1] https://github.com/IntelAI/dynamic-reparameterization
> [2] https://arxiv.org/abs/1904.01569
> [3] https://arxiv.org/abs/1810.02340
> [4] https://openreview.net/forum?id=B1VZqjAcYX
> [5] https://github.com/gahaalt/SNIP-pruning

---

> ### Author Response · Authors · 2019-11-15
> **Dense performance for ImageNet**
>
> We calculated at which sparsity level we reach dense performance. We ran 3 experiments on the dense ImageNet variant based on the Dynamic Sparse Reparameterization codebase (which is used to replicate all other methods as well). We find that for accuracy the ImageNet performance is 75.31% with 95% confidence interval in (74.91, 75.71). Sparse momentum with 50% weights scores 74.94% accuracy. As such, it is reasonable to say that we reach dense performance with 50% weights.
>
> Furthermore, we discussed [1] in the comment above (Title: "Results from learning rate grid search")  and will include this discussion in our paper.
>
> We kindly request a response from you before the rebuttal period ends. Your review will determine the outcome of this review process. It is frustrating to us that your review contains what we believe to be misconceptions but you are not willing to start a dialog to settle these concerns. We cannot defend ourselves against your assessment if you do not respond in a timely manner.

---

### Official Review · AnonReviewer2 · 2019-10-24
**Official Blind Review #2**

**Rating:** 6

**Review:**

This paper proposes an algorithm called Sparse Momentum for learning sparse neural networks. They claim to maintain sparse weights throughout training while achieving dense network performance levels. They also show their method improves training speed up to 5.61x faster training. The provides a decent motivation for why sparse networks can be helpful. The related work section is well summarized and they emphasize that the current work's primary motivation is to reduce training time while maintaining performance. They compare their method with other methods that also maintain sparse neural networks throughout training and involve single training phase, which is fair. Their method consists of primarily 3 phases - i) pruning weights ii) redistribution of weights iii) regrowing of weights based on the exponentially smoothed momentum term for each layer. The method is well explained and motivation is clear. Edge case was also explained for more clarity. However, there is something that needs more clarification in Pg-3, 3rd line, they say the 3 components of the algorithm i) ii) and iii) can be tackled independently with a divide and conquer strategy to gain some computational benefits. In my understanding, the method performs these 3 steps sequentially after each epoch. Not sure how divide and conquer strategy can be used here ?

The results were shown for MNIST & CIFAR-10 using AlexNet, VGG16 and LeNet-5 models. They show that the current algorithm is better than other proposed methods in the literature in most of the cases. The claims made in some cases that this method reaches dense network performance is not completely true though. For example, in Table 1, the only case where the proposed method reaches dense network's performance is for VGG16-D. In all the rest of the cases, the current method and dense network error differ by at least 5%.

The authors compare speed up results for training in two ways: theoretical speedups which are proportional to reduction in number of FLOPS and practical speedups using dense convolutional algorithms corresponding to completely empty channels. It's good that the authors have mentioned due to the current lack of optimal sparse matrix multiplication implementations these speedups cannot be garnered practically yet. The estimates based on FLOPs reduction or Empty Channel based look promising. They also show ablation study of how the redistribution and weight regrowth based on momentum is better than doing in a random fashion.

Overall, I think the paper proposes an interesting idea of using momentum with promising results to learn sparse neural networks.

**Experience Assessment:**

I have read many papers in this area.

**Review Assessment: Checking Correctness Of Derivations And Theory:**

I assessed the sensibility of the derivations and theory.

**Review Assessment: Checking Correctness Of Experiments:**

I assessed the sensibility of the experiments.

**Review Assessment: Thoroughness In Paper Reading:**

I read the paper at least twice and used my best judgement in assessing the paper.

---

> ### Author Response · Authors · 2019-11-06
> **Clarifications**
>
> Question: “[...] components of the algorithm i) ii) and iii) can be tackled independently with a divide and conquer strategy to gain some computational benefits. In my understanding, the method performs these 3 steps sequentially after each epoch. Not sure how divide and conquer strategy can be used here ?”
>
> Answer: Thank you for your feedback! Sorry, using divide-and-conquer in this context was an unfortunate word choice. We meant the meaning of this term not in the computer science sense, but in the general sense: To solve a problem by separating it into sub-problems which are solved independently.
> What we mean by this, is that we separate the problem of sparse learning into (i) pruning, (ii) growing, (iii) redistribution and we solve these problems one-by-one by using heuristics (statistics such as the mean magnitude of the momentum vector). Our interpretation of these components as independent building blocks might be more apparent from our ablation analysis, where we exchange these components with components from other methods such as SET and DSR.
> Other methods in the literature like Deep-R try to solve all these problems all at once by using a Bayesian method which is expensive to compute, whereas our heuristics are fast to compute and simple. This was the meaning of the sentence: “ In this work, we follow a divide-and-conquer strategy that is guided by computationally efficient heuristics.“
> We will rewrite this sentence and parts of this section to prevent this confusion.
>
>
> Question: “The claims made in some cases that this method reaches dense network performance is not completely true though. For example, in Table 1, the only case where the proposed method reaches dense network's performance is for VGG16-D. In all the rest of the cases, the current method and dense network error differ by at least 5%. “
>
> Answer: Table 1 was not meant to represent our experiments to replicate dense performance levels. Dense performance levels are presented in Table 3. We will remove mentions in Table 1 and hint at this in the introduction to prevent this misunderstanding.
> Furthermore, dense performance level equivalents for ImageNet are missing and we will produce these numbers and will append them to the appendix. Please stay tuned for the updated results.

---

### Official Review · AnonReviewer1 · 2019-10-27
**Official Blind Review #1**

**Rating:** 6

**Review:**

In this paper, the authors propose a sparse momentum algorithm for doing efficient sparse training. The technique relies on identifying weights in a layer that do not have an effect on the error, pruning them, and redistributing and growing them across layers. The technique is compared against other recent algorithms on a range of models.

The paper is well written, and very easy to read. The proposed algorithm looks interesting and seems to empirically work well.

I am, however, a bit confused with how the sparse momentum algorithm is discussed in this paper. In the paper, momentum is intuitively presented as an algorithm that reduces the variance of the noise in the gradients. However, there are a number of papers that show that this is not the case (and if anything it is the opposite). Further, a few recent papers show that momentum works better than SGD only for learning rates that are not too small, and this is because it averages out the gradients in the high-curvature directions (these are the directions where the gradients switch signs) and makes them stable in these directions, thus allowing larger steps in the low curvature directions. See for example the following two papers:
Momentum Enables Large Batch Training. Samuel L Smith, Erich Elsen, Soham De. ICML Workshop on Physics for Deep Learning, 2019.
Which Algorithmic Choices Matter at Which Batch Sizes? Insights From a Noisy Quadratic Model. Guodong Zhang, Lala Li, Zachary Nado, James Martens, Sushant Sachdeva, George E Dahl, Christopher J Shallue, Roger Grosse. NeurIPS 2019.

Given these papers, can the authors comment on what they think is the reason for the effectiveness of their sparse momentum algorithm? These papers seem to indicate to me that an interesting (and important) ablation study would be to compare using just the gradients vs using momentum for the sparse training algorithm for both a small batch and a large batch. Without this ablation study, it is a bit unclear to me why/when this algorithm is working well, and this primarily explains my current score.

The experimental results look impressive, although I am not very aware of other work in sparse training, so unfortunately it is harder for me to properly evaluate the significance of the empirical results in this paper, and whether the numbers reported are indeed a significant improvement over current state-of-the-art algorithms. I do have a few questions about the design and the results of the experiments presented:

1. Because the momentum of zero-valued weights are used, does that mean that the gradient over all weights are required to be taken at each step? How much does this affect training time in your experiments?

2. How were the learning rates decided, and why are they kept fixed across methods? It seems feasible that the optimal learning rate could vary highly between methods?

==========================================

Edit after rebuttal:
I thank the authors for the very detailed response and for doing the additional experiments requested. Due to the thoroughness of the response, I am increasing my score to a weak accept. I do however have a couple of comments regarding the author response:

1. I do not agree that keeping the learning rate fixed across methods is the right approach. Many different things might interact with each other to change the optimal learning rate, and it is always more interesting to see the optimal performance attained by a method through a grid search. As long as the tuning budget across all methods are somewhat similar, I think the experiments would be more informative when doing a learning rate sweep. Further, while doing the sweep, the authors should be careful that the optimal learning rate does not lie on the edge of the grid search. This is what happens in the experiments reported in the authors' response titled "Results from learning rate grid search" where the optimal learning rate for all models lie at one of the extremes of the grid search (0.09).

2. I would request the authors to slightly rewrite certain parts of their paper so as not to imply that momentum decreases the variance of the gradients in general. There are many papers (including the ones I mentioned in my review) that specifically show momentum does not have this effect. All of these papers however consider a different setting from the one considered in this paper (where there is a parameter redistribution step), so it is not immediately clear whether momentum has the same effect in this case. However, unless there is very specific evidence to show a variance reduction effect, I think putting in that intuition might be misleading.
It is interesting to see that momentum helps for small batches in this setting, and I agree that further investigation into this would be an interesting future direction. Note that there is a concurrent submission on sparse training (https://openreview.net/forum?id=ryg7vA4tPB&noteId=ryg7vA4tPB) that seems to show that higher momentum values (0.99) does better when considering large batches. So perhaps additional and more careful experiments need to be done about this before these additional experimental results can be put in the paper.


**Experience Assessment:**

I have read many papers in this area.

**Review Assessment: Checking Correctness Of Derivations And Theory:**

I assessed the sensibility of the derivations and theory.

**Review Assessment: Checking Correctness Of Experiments:**

I carefully checked the experiments.

**Review Assessment: Thoroughness In Paper Reading:**

I read the paper at least twice and used my best judgement in assessing the paper.

---

> ### Author Response · Authors · 2019-11-07
> **Addressing intuition about momentum and other concerns.**
>
> Regarding the intuition about momentum
> ===================================
>
> Thank you for raising the points about the interpretation of momentum in our procedure. We were thinking about this in more detail, but due to space considerations we opted for a more intuitive explanation rather than providing a more rigorous analysis — after all, the main contribution of our work is to replicate dense performance with faster training time rather than giving a rigorous analysis of why momentum works in this case.
>
> The thinking of why momentum is better than gradients has two reasons. First, the more simplistic reason: because we do parameter redistribution once per epoch (instead of every gradient step), the last gradient before we perform redistribution is a poor estimate for how the error landscape evolves during training. If I have gradients 0..100 and I just take the 100th gradient to make estimates about redistribution, then the parameter redistribution is usually poor. This intuition corresponds to our unreported experimental results on gradients vs momentum. You asked for an analysis of this and we will provide results in due course (for both small and large batch sizes) and add them to the appendix.
>
> The second reason is that recent work suggests that the optimization of neural networks happens in a tiny subspace [1,2] that corresponds to the directions spanned by the Hessian eigenvectors with the largest eigenvalues. However, to yield good predictive performance, it is important to optimize along the eigenvectors with small eigenvalues. This intuition corresponds to your intuition and the work that you cite, which finds that momentum allows the optimization of low curvature directions. The authors of [2] also find optimization along low curvature directions is also the reason why batch normalization and learning rate schedules are important for low good generalization.
> Thus similarly, in our case, we believe that using momentum instead of gradients in redistribution facilitates optimization along directions with low curvature. We verified this belief experimentally for small batch sizes (128) and a standard learning rate + learning rate schedule (0.01, drop ever 40% of training epochs by one order of magnitude). From these initial results on small batch sizes, it seems the learning rate of 0.01 is large enough for momentum to be beneficial over SGDs.
>
> Please let us know if you have further concerns regarding our intuition.
>
> We think it is unfitting to report this intuition in our work without rigorous evidence that supports it. What kind of change to our work would you recommend to incorporate our intuition?
>
>
> Other Concerns
> =================================
>
> Question: "Because the momentum of zero-valued weights are used, does that mean that the gradient over all weights are required to be taken at each step? How much does this affect training time in your experiments?"
>
> Answer: Because the momentum of zero-valued weights needs to be calculated we have dense weight gradients. However, we still use sparse deltas (errors from the layer above, passing the error along the network) which make up the largest chunk of computation in convolutional networks. To estimate by how much time is spent on computing the dense weight gradients, we measure the median computational overhead of dense weight gradient computation on CIFAR-10 for various input/output/weight sizes. We find that the dense weight gradients to make up 2.6% of the overall computation in a forward/backward pass. Thus having dense weight gradients is computationally negligible.
>
> Question: "How were the learning rates decided, and why are they kept fixed across methods? It seems feasible that the optimal learning rate could vary highly between methods?"
>
> Answer: The learning rate question is a dilemma we have been thinking hard about: We build on work (SNIP, SET/Deep-R/DSR) which compares across methods by using a fixed learning rate schedule and a fixed learning rate. This has the advantage that it introduces a fixed bias and it makes it easier to replicate results.
> However, it is probably right, that our method, or indeed other methods, could be improved by simply searching in the learning rate space. The issue with that is, that one can quickly gain an unfair advantage over other methods. It is for example known, that learning rate schedules with warmups perform significantly better than learning rate schedules without warmup.
> A fair comparison would be to not alter the learning rate schedule and just search the learning rate space with a grid search. We will do this together with the analysis above and provide results in the appendix.
>
> [1]: Gradient Descent Happens in a Tiny Subspace: https://arxiv.org/abs/1812.04754
> [2]: An Investigation into Neural Net Optimization via Hessian Eigenvalue Density: https://arxiv.org/abs/1901.10159

---

> ### Author Response · Authors · 2019-11-07
> **Results from learning rate grid search**
>
> Here the results of a grid search for the learning rate in {0.001, 0.003 , 0.006, 0.01, 0.03, 0.06, 0.09} while keeping the batch size fixed (128). We perform this grid search on AlexNet-b, VGG-D, and WRN 16-8 to provide one model of each class of models. We find that larger learning rates work better which is in line with findings from [1] for other sparse methods.
>
> This suggests in combination with [1], that varying larger learning rates, rather than keeping them fixed across methods, would give an unfair advantage to sparse methods that tune the learning rate. This justifies reporting the results for a fixed learning rate of 0.01.
>
> Note that we spent roughly 400 GPU hours on this grid search for 3 models and a single method. Running all these experiments with a grid search on the learning rate for each method on CIFAR-10 alone would easily exceed a computational budget for 8000 GPU hours which would be 82 days of 24/7 experiments with a 4 GPU system. Under these considerations, we argue that keeping a fixed learning rate for each method and reporting optimal learning rates for our own method in the appendix is the best we can do. Demands for a learning rate search across methods is computationally unreasonable.
>
> Results:
>
> Best learning rate for each model
> model	batch size	lr	       test acc	   SE       test error	    method
> alexnet-b	128	    0.09	        86.507	0.1386	13.493	sparse momentum
> vgg-d	        128	    0.09	        93.648	0.1081	  6.352	sparse momentum
> wrn-16-8	128	    0.09	        94.550	0.0187	  5.450	sparse momentum
>
>
> [1] Rethinking the Value of Network Pruning: https://arxiv.org/abs/1810.05270v2

---

> ### Author Response · Authors · 2019-11-08
> **Ablation results for large/small batch size for gradient/momentum methods.**
>
> Here the ablation study results where we compare momentum growth and redistribution with gradient growth and redistribution. We present results for small batch size (64) and a large batch size (1024). We search the best learning rate in the interval {0.001, 0.003 , 0.006, 0.01, 0.03, 0.06, 0.1, 0.3, 0.6, 0.9} and report the best results for each method (momentum vs gradient).
> Overall, we find that momentum is superior in the low-batch size regime, but the plain gradient does better for large batch sizes. These results are in direct opposition to your source that you cite [1].  Furthermore, we find that the gradient method is by far inferior to our momentum method as it does not come close to the performance of sparse momentum. We will include these results in our appendix.
>
> This confirms our intuition that momentum is superior to the mere gradient because, firstly, the gradient just captures a single point in time while the momentum captures an exponentially weighted average of the optimization process. Thus making a redistribution and growth decision based on the information of single gradient yields sub-optimal results.
> Secondly, the results accord with your intuition that momentum might help optimization along low-curvature directions. The results contradict that momentum methods are only effective with large batch sizes.
>
> We believe these results bolster our claims and intuition and further investigation on this matter is out of scope of our work. We did our due share. Further investigation should be part of a separate paper that investigates interpretations of momentum on the optimization process directly.
>
> Results:
>
> Best model for large batch size
> ----------------------------------------------
> model	batch size	lr	  test acc	SE	  error	method
> alexnet-b	1024	0.06	  82.983	0.4268	17.017	momentum
> vgg-d	        1024	0.03	  91.508	0.1968	8.492	momentum
> wrn-16-8	1024	0.09  92.875	0.0437	7.125	momentum
> ---------------------------------------------------------------------------------------------
> alexnet-b	1024	0.6	  84.505	0.455	15.495	gradient
> vgg-d	        1024	0.09	  91.663	0.0273	8.337	gradient
> wrn-16-8	1024	0.6	  94.085	0.035	5.915	gradient
>
> Best model for small batch size
> -----------------------------------
> model	batch size	lr	test acc	  SE	        error	method
> alexnet-b	64	     0.03	86.885	0.122	13.115	momentum
> vgg-d	        64	     0.03	93.625	0.0779	6.375	momentum
> wrn-16-8	64	     0.09	94.38	0.0464	5.62	        momentum
> ---------------------------------------------------------------------------------------------
> alexnet-b	64	     0.06	86.567	0.1545	13.433	gradient
> vgg-d	        64	     0.06	93.533	0.109	6.467	gradient
> wrn-16-8	64	     0.03	94.15	0.04	        5.85	        gradient
>
> [1] Momentum Enables Large Batch Training. Samuel L Smith, Erich Elsen, Soham De. ICML Workshop on Physics for Deep Learning, 2019.

---

### Decision · Program_Chairs · 2019-12-19

**Decision:**

Reject

**Comment:**

This paper presents a method for training sparse neural networks that also provides a speedup during training, in contrast to methods for training sparse networks which train dense networks (at normal speed) and then prune weights.

The method provides modest theoretical speedups during training, never measured in wallclock time.   The authors improved their paper considerably in response to the reviews.  I would be inclined to accept this paper despite not being a big win empirically, however a couple points of sloppiness pointed out (and maintained post-rebuttal) by R1 tip the balance to reject, in my opinion.  Specifically:

1) "I do not agree that keeping the learning rate fixed across methods is the right approach."  This seems like a major problem with the experiments to me.

2) "I would request the authors to slightly rewrite certain parts of their paper so as not to imply that momentum decreases the variance of the gradients in general."  I agree.